# Hypothesis- and Structure-based prompting for medical and business diagnosis

## Abstract

In real-world scenarios like healthcare and business, tackling 'many-to-one' problems is challenging but crucial. Take medical diagnosis: A patient's chief complaint can be caused by various diseases, yet time and resource constraints make identifying the cause via difficult. To tackle these issues, our study introduces Hypothesis-based and Structure-based (HS) prompting, a method designed to enhance the problem-solving capabilities of Large Language Models (LLMs). Our approach starts by efficiently breaking down the problem space using a Mutually Exclusive and Collectively Exhaustive (MECE) framework. Armed with this structure, LLMs generate, prioritize, and validate hypotheses through targeted questioning and data collection. The ability to ask the right questions is crucial for pinpointing the root cause of a problem accurately. We provide an easy-to-follow guide for crafting examples, enabling users to develop tailored HS prompts for specific tasks. We validate our method through diverse case studies in business consulting and medical diagnosis, which are further evaluated by domain experts. Interestingly, adding one sentence "You can request one data in each response if needed" initiates human interaction and improves performance.

## 1 Introduction

Real-world problems often exhibit many-to-one relationships, where multiple factors can lead to a single observed outcome. This is prevalent in domains like medical diagnosis, where a range of underlying diseases can manifest similar symptoms, as well as in business, where declining sales might be attributed to either new competitors or deteriorating product quality. Identifying the root cause is paramount as it sets the foundation for devising appropriate solutions; a treatment plan in medical cases or a strategic adjustment in a business context. However, the straightforward computerized approach for identifying root causes–collecting comprehensive data and conducting exhaustive searches–is often untenable due to the vastness of the search space and limitations on resources such as time and budget. For instance, there is a healthcare scenario where a 30-years old woman patient arrives with a high fever. Physicians are tasked with narrowing down an array of possible underlying causes by judiciously selecting tests and examinations. However, they can't gather every piece of data, as some tests are time-consuming and may delay critical treatment for conditions like fulminant bacterial infections.

LLMs have shown strong performance in both general tasks (Qin et al., 2023) and real-world modeling (Zheng et al., 2023) by utilizing vast knowledge acquired from extensive web data. However, mere possession of domain-specific knowledge is insufficient to effectively navigate these complex spaces. LLMs need to apply this knowledge in a structured and efficient manner, especially when solving many-to-one problems. The key lies in guiding LLMs effectively through specialized prompting techniques that take into account the inherent complexities of these situations. Unfortunately, previous prompting methods, such as Chain-of-Thought (CoT) (Wei et al., 2022; Creswell et al., 2022; Lewkowycz et al., 2022; Wang et al., 2022), Tree-of-Thought (ToT) (Yao et al., 2023), and Graph of Thoughts(GoT) (Besta et al., 2023) have limitations in addressing these challenges. While CoT excels in one-to-one mapping problems, it falters when multiple potential root causes must be explored. ToT and GoT, though adept at simpler recursive problem-solving tasks, are limited to relatively simple and well-defined repetitive tasks–such as sorting and crosswords puzzle–which makes it less applicable to real-world complex challenges such as medical or business diagnosis.

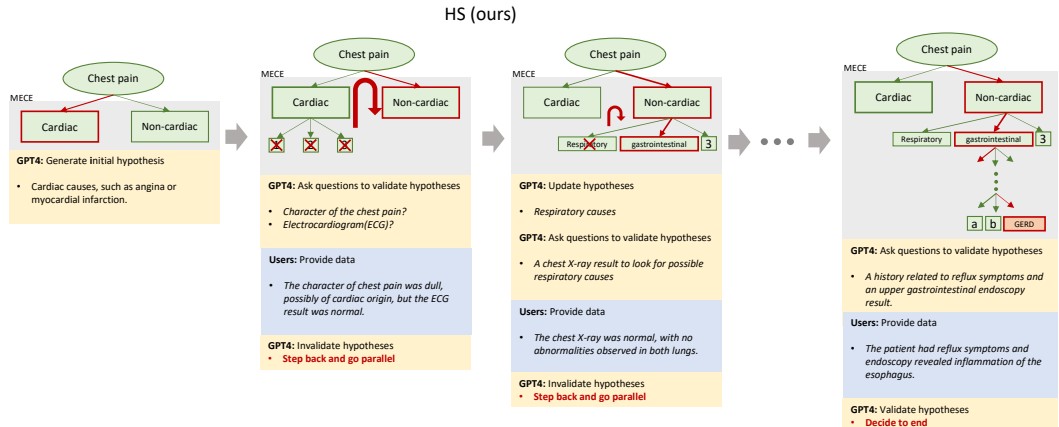

Figure 1: Overview of the HS framework using a medical example. A 37-year-old woman reports chest pain. The HS method systematically categorizes the potential causes into: (a) Cardiac and (b) Non-cardiac. Leveraging this structured approach, the model formulates and validates hypotheses through targeted data inquiries. Within a few iterations, the root cause, GERD, is accurately identified. Further details can be found in Section 3.

To address these challenges, we introduce a Hypothesis-based and Structure-based (HS) method, specifically engineered for tackling many-to-one problems. This approach uniquely integrates a structure-based methodology, using the Mutually Exclusive, Collectively Exhaustive (MECE) principle to break down complex problems into discrete, digestible parts. For example, physicians can segregate possible causes of a fever into broader categories with structure suitable for the domain like infectious and non-infectious diseases or emergency and non-emergency. Guided by this structure, the model generates and prioritizes hypotheses derived from available data. To confirm or reject these hypotheses, the HS method encourages the model to actively seek additional information through well-crafted queries. Importantly, our method allows the model to understand its current position within the overall problem-solving framework. This holistic understanding helps the model to strategize its next moves effectively, making the entire diagnostic process not just efficient but also highly targeted. We provide an easy-to-follow guide for crafting examples, enabling users to develop tailored HS prompts for specific tasks.

To validate our approach, we assess LLMs equipped with various prompting methods across three consulting cases and four medical diagnostic scenarios. These cases span a broad spectrum of "many-to-one" problems in both the business and healthcare sectors. Our results, evaluated by a panel of five experienced consultants and five medical doctors, reveal both the strengths and limitations of our HS prompting method in addressing complex, real-world challenges. Interestingly, we found out that one phrase "You can request one data in each response if needed" improve performance effectively, similar to "Let's think step by step" in Kojima et al. (2022).

We highlight the following:

- We propose a unique prompting structure, HS, that encourages models to employ structure-based and hypothesis-based thinking in their problem-solving process for various many-to-one problems in real-world. Our method guides LLMs to utilize a MECE framework, prioritize hypotheses, and actively inquire to verify these hypotheses.

- We offer an easy-to-follow guide for generating examples, allowing users to create appropriate examples tailored to their specific tasks. By aligning the examples with our structure-based and hypothesis-based approach, users can stimulate the LLMs to solve problems more effectively and efficiently.

- We enlist domain experts in business consulting and medical diagnosis to validate our method, providing a comparison to existing baseline methods. This direct involvement of seasoned professionals ensures our approach's practical applicability and credibility.

## 2 RELATED WORK

**Chain-of-Thought and Self-reflection** The Chain-of-Thought (CoT) method (Wei et al., 2022) and its subsequent refinements (Creswell et al., 2022; Lewkowycz et al., 2022; Wang et al., 2022) have been effective in solving problems that involve a straightforward one-to-one mapping between problem and answer. Some even incorporate a magic sentence like 'Let's think step by step' (Kojima et al., 2022) to enhance problem-solving. Meanwhile, self-reflection techniques (Paul et al., 2023; Shinn et al., 2023; Madaan et al., 2023) focus on improving model outcomes by iteratively reviewing and adjusting generated responses. However, these methods often lack the ability to explore multiple options for solutions in a structured way, thereby missing potential root causes. Furthermore, prompts using majority vote like self-consistency (Wang et al., 2022), ask-me-anything (Arora et al., 2022) are not effective in complex real-world tasks, where final answers may be largely different based on which routes the model takes. Another noteworthy difference is that the original CoT paper (Wei et al., 2022) and subsequent works usually rely on eight manually-crafted examples for their prompts, we utilize a single example that aligns with our five-step guide for problem-solving.

**LLMs on complex tasks** The work by Dziri et al. (2023) looks at how LLMs handle simple tasks, such as multiplying numbers, and explores whether this can extend to more complicated problems. The study by Zhou et al. (2022) goes deeper into the reasoning process, splitting it into parts like defining the problem and finding solutions. Some techniques, like the ones from Lightman et al. (2023); Uesato et al. (2022), make it easier for LLMs to deal with complex tasks by breaking them down into smaller steps with rewards. Multi-step reasoning prompting methods, such as Tree-of-Thoughts (ToT) (Yao et al., 2023), Graph-of-Thoughts (GOT) (Besta et al., 2023), and Reasoning-via-Planning (RAP) (Hao et al., 2023) employ graph search algorithms outside LLMs to generate and select options efficiently. Self-eval guided decoding (Xie et al., 2023) integrates self-evaluation to guide the beam searching process. Detailed comparison of ToT, GoT and ours(HS) can be found in the Section 5.

**LLMs in Medical Applications** In the realm of medical question-answering tasks, such as MedQA (USMLE) (Jin et al., 2021), and PubMedQA (Jin et al., 2019), LLMs such as GPT-3 (Brown et al., 2020) and Flan-PaLM (Chowdhery et al., 2022; Chung et al., 2022) have made substantial strides by training on large-scale internet corpora. Beyond question-answering, GPT-3 has demonstrated significant capabilities in various medical subfields including diagnosis, surgery, genetics (Levine et al., 2023; Duong & Solomon, 2023; Oh et al., 2023). Building on strong baseline LLMs, Ayers et al. (2023) compared the responses of ChatGPT and physicians to patient questions sourced from a social media forum. Med-PaLM (Singhal et al., 2023a) and Med-PaLM2 (Singhal et al., 2023b) examined the performance of fine-tuned PaLM (Chowdhery et al., 2022) and PaLM2 (Anil et al., 2023) models in multiple-choice medical benchmarks and long-form question answering. These LLMs received higher ratings for both quality and empathy. Nonetheless, these studies do not explore LLMs' capabilities in active, interactive medical diagnosis settings. The previous models were evaluated using multiple-choice questions, which present all information simultaneously for decision-making or simple medical question-answering tasks. In contrast, our study assesses LLMs in a more dynamic scenario where they must actively inquire for additional patient data to accurately diagnose conditions—mimicking real-world medical practice more closely.

We discuss additional related work including LLMs in business applications in the Appendix.

## 3 PROPOSED METHOD

The HS method seamlessly integrates structures with hypothesis-driven reasoning, equipping LLMs with the tools to effectively tackle complex many-to-one problems. Our approach begins with a thorough and efficient exploration of the problem space through structured decomposition. Then, LLMs generate hypotheses in line with this structure, prioritizing them based on available data and their likelihood of being correct. A crucial element of our prompting method is that it encourages LLMs to ask sharply targeted questions or request specific data to validate or invalidate these hypotheses. The capability to ask the right questions is vital for arriving at accurate root cause of the problem.

We provide an easy-to-follow guide for generating examples, allowing users to create appropriate examples tailored to their specific tasks. Here are details about five steps:

**1. Problem Definition** The process initiates by requiring the LLMs to clarify the client's objectives and their current state clearly. This step not only ensures that the model comprehends the task at hand, but it also guides its subsequent problem-solving efforts in alignment with the users' goals. For example, if the client's goal is framed as 'becoming a market leader', the LLM is prompted to seek clarification by asking questions to determine whether 'market leadership' is defined by revenue, market share, or another key performance indicator.

**2. Structuring the Problem** Upon securing a comprehensive understanding of the problem, the LLMs are directed to create a structure of the problem landscape, rooted in the MECE principle. For instance, in a scenario where increasing revenue is the goal, the problem could be bifurcated into (a) broadening the customer base and (b) amplifying average revenue per customer. Each of these major categories can be further dissected into MECE sub-categories.

**3. Hypothesis Generation** With the structured framework in place, the LLM is well-positioned to proceed to hypothesis generation. Within the confines of this structured problem layout and informed by the task's specifics, the LLM generates a set of hypotheses. The crucial step here is the prioritization of these hypotheses, determined by factors such as their likelihood based on available data and their potential impact on solving the problem. This prioritization acts as a navigational aid, thereby guiding the model through its subsequent problem-solving stages.

**4. Efficient Search Process** To validate its prioritized hypotheses, the LLM engages in an active and strategic search within the framework. Central to this phase is the model's ability to solicit targeted questions or request specific data that are crucial for hypothesis validation. Upon receiving and analyzing the data, the model performs a self-evaluation of its current hypotheses. It then chooses among several courses of action based on this assessment: 1) Stop and formulate a solution if the current node provides a holistic and detailed resolution to the problem. 2) Go down the tree if the data confirms the validity of the current node's hypothesis. 3) Go parallel to investigate alternative, more plausible nodes if the data calls the current hypothesis into question. 4) Go up (step-back) in the hierarchy if the data fails to validate any nodes at the current depth level. 5) Change the whole framework if it becomes evident that a solution cannot be reached within the current structure.

**5. Develop Solution** Lastly, based on the selected hypotheses pinpointing the root causes of the problem, the model proposes solutions while considering potential risks and uncertainties.

Here, we give medical diagnosis illustrative example following HS framework in Figure 1.

***Problem definition and structure***: A 37-year-old woman presents with about 2 weeks of chest pain. The problem is framed in a MECE manner, with two first-level categories: 1) Cardiac causes and 2) Non-cardiac causes.

***Hypothesis generation:*** The model initially hypothesizes that the chest pain is due to a cardiac issue, such as angina or myocardial infarction.

***Efficient Search Process:***
***Step 1)*** To validate this hypothesis, the model requests two pieces of initial data: 1) A nature and character of the chest pain, and 2) A Electrocardiogram(ECG) result to look for an evidence of cardiac diseases. The character of chest pain was dull, possibly of cardiac origin, but the ECG result was normal. Given the conflicting data, the model opts to "go parallel" to explore the other initial MECE option—Non-cardiac causes. It requests past history and recent medical history to investigate.
***Step 2)*** The patient recently experienced upper respiratory tract symptoms, which leads the model to refine the hypothesis of non-cardiac issue, and decide to "go down the tree" to explore a new hypothesis: "The cause of chest pain is likely a respiratory origin." It requests a chest X-ray result to look for possible respiratory causes.
***Step 3)*** The chest X-ray was normal, with no abnormalities observed in both lung. Faced with this conflicting information, the model decides to "step back" and reevaluate its previous hypotheses. It considers that the chest pain could be gastrointestinal origin and requests a history related to reflux

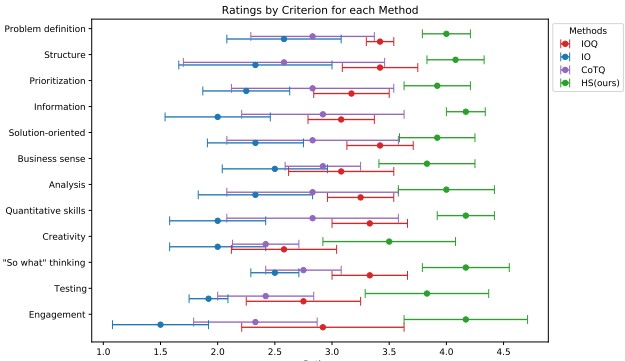

Figure 2: Median and quartiles for each criterion in the business domain, averaged across all cases based on different prompting methods.

symptoms and an upper gastrointestinal endoscopy result.

***Step 4)*** The patient had recently been experiencing reflux symptoms and endoscopy revealed inflammation of the esophagus, which is suggestive of gastroesophageal reflux disease (GERD). Since this new hypothesis aligns well with the new data, the model decides to "terminate the search", concluding that GERD is the detailed and holistic root cause of the patient's chest pain.

***Develop Solution*** Based on the efficient search and validation process, the model recommends antacid treatment and lifestyle modifications for treating GERD, which is diagnosed as the root cause of the patient's chest pain.

HS prompts with detailed business and medical examples are in the Appendix.

## 4 RESULTS

**Model and Metrics** For the research presented in this paper, we focused exclusively on the GPT-4 provided by OpenAI's chat interface. Note that our usage of GPT-4 was specific to certain periods: for business cases, we engaged with the model from August 10 to 20, while for medical cases, we utilized it from September 10 to 20. We assessed each case using the median score and the 25% and 75% quartiles to offer further insight into score distribution, a common approach in survey analysis.

**Baselines** Our baselines are IO, IOQ, and CoTQ. We use the term 'Q prompting' when we enhance existing prompting methods like IO and CoT with an added instruction: "You can request one [data/clinical information] in each response." This encourages the LLMs to initiate interaction with users. We evaluated CoT's efficacy using a challenging medical case for an ablation study. We observed that complex, multi-step reasoning methods, such as ToT Yao et al. (2023), GoT Besta et al. (2023), and RAP Hao et al. (2023), don't align well with interactive, real-world scenarios like business consulting and medical diagnosis. While these methods utilize the GPT-4 API to integrate tree or graph search algorithms, these methods best suit tasks that can be broken down into repetitive steps, with a clearly defined endpoint. However, the non-linear and undefined nature of steps in our tasks makes it challenging to determine how and when to effectively integrate search algorithms outside LLMs in these baselines. Furthermore, our research involves interactive scenarios where the user provides data in response to the model's queries. This dynamic interaction complicates the direct application of baseline methods that rely on predefined search algorithms asking GPT-4 for options and making selections.

### 4.1 BUSINESS CONSULTING

**Business consulting case selection** This study delved into the analytical challenges of diagnosing the root causes of decreasing profitability in business scenarios. We curated a set of three business cases, referring to the renowned Kellogg Business Case Book and Interview Guide Carbon Dioxide Research Group (2004), avoiding data leakage problems. These cases were structured to emulate

the diagnostic challenges inherent in identifying profitability decline of companies, reflecting real-world business complexities. Our case selection was based on the following criteria: 1) We selected cases with different domains and industries such as food product, franchise restaurant, and insurance business. 2) We focused on cases about profitability decline because they have clear root causes. This helps us better test the diagnostic skills of the methods in our study compared to cases on market entry or marketing strategies. 3) We chose cases by the complexity of diagnosis. Some cases have hidden root causes, while others are clearer. Details of cases can be found in Appendix.

**Criterion** Since there are no official fixed-form evaluation criteria for business consulting cases, we refer to the Kellogg MBA consulting club case book and check the validity of them from three management consultants from McKinsey and Deloitte. To streamline our evaluation parameters, we started with a set of 30 potential criteria, which was suggested in the Kellogg MBA consulting club case book. Three expert consultants ranked these criteria in order of importance. Alongside this, they provided a binary mask for each criterion to indicate its necessity. By merging the rank and the binary feedback, we were able to identify and finalize 12 essential criteria for the assessment. We confirm that experts who set the criteria were not involved in the scoring process. Detailed criteria can be found in the Appendix.

**Evaluators** In our evaluation process, we engaged five business consultants, each either holding an MBA or possessing over five years of experience from reputable consulting firms. Their primary task was to evaluate reports generated by the GPT-4 model spanning 26 cases, encompassing three distinct cases and four baseline methods. To elaborate, we produced three trials for each of the four baseline methods applied to the three cases. From the aggregate of 36 trials, two consultants selected both the best and the worst trials for each method. This resulted in a selection of 24 trials for the primary study, which were subsequently presented to human experts. With the inclusion of five consultants in our evaluation process, case 1 was evaluated by five consultants, and case 2 and case 3 were evaluated by four consultants. Our conclusive report delineates the average scores assigned to each method and further emphasizes the degree of consensus among the consultants regarding these scores. Additionally, we get interviews for qualitative analysis.

|        | IO                | IOQ                    | CoTQ              | HS(ours)              |
|--------|-------------------|-----------------------|-------------------|-----------------------|
| Case 1 | 3.04 [2.46, 3.54] | **3.79 [3.58, 4.17]** | 2.08 [1.71, 3.12] | 3.33 [2.79, 3.79]     |
| Case 2 | 1.81 [1.49, 2.12] | 2.88 [2.54, 3.20]     | 2.90 [2.54, 3.32] | **4.58 [4.20, 4.84]** |
| Case 3 | 1.71 [1.38, 2.15] | 2.77 [2.49, 3.19]     | 3.15 [2.81, 3.43] | **4.02 [3.54, 4.40]** |
| Avg    | 2.19 [1.78, 2.60] | 3.15 [2.87, 3.52]     | 2.71 [2.35, 3.29] | **3.98 [3.51, 4.34]** |

Table 1: Median and quartiles for each business case, averaged across all criteria based on different prompting methods.

**Results** As presented in Table 2, our HS method leads in terms of average score, holding a gap of 0.83 points above the next best performer, the IOQ method. On closer inspection, the HS method consistently scores above its peers across all 12 criteria, thereby achieving the highest overall rating. The efficacy of the HS method can be traced back to its organized tree approach combined with the adaptive adjustment of hypotheses through proactive information retrieval (further detailed in the Qualitative analysis section below).

When breaking down the results by individual cases, HS slightly lags behind in case 1, with IOQ taking the lead, yet still surpasses CoTQ in terms of average rating. For case 1, all methods scored relatively low, as none could precisely identify the core issue: a decline in profitability. More specifically, HS did not delve deep enough, settling for a surface-level explanation due to its confined self-evaluation capabilities. In contrast, other methods struggled to generate a proper structure with MECE principle, thus overlooking key analytical perspectives. Further insights on this are elaborated in Section 5. In cases 2 and 3, HS effectively worked through the necessary analytical dimensions. It pinpointed the root cause by splitting the issue into revenue and cost components and then further explored the cost-related challenges. This thorough analysis earned HS a commendable evaluator rating of over 4. In contrast, both CoTQ and IOQ, without a structured approach, only grazed the problem's surface. They didn't identify the root cause even after multiple data requests.

When reviewing performance across different criteria, the most noticeable areas (with the top 5 differences) where HS surpassed the second best, IOQ, were engagement, information, testing, cre-

ativity, and 'So what' thinking. This performance edge is linked to our prompts that nudged it to handle the case based on hypotheses and a structured framework. We delve deeper into this observation with insights from interviews with other consultants in the following section.

**Qualitative analysis: what makes HS better** The enhanced performance of HS method compared to the next best scorer, IOQ, and other methods in case 2 and case 3 shows the importance of the structure basis approach is critical for its problem-solving performance. A closer look reveals the following insights. First, HS employs a structured framework that respects the MECE principle. For instance, in case 2, HS first partitioned the problem into revenue and cost, then zoomed into variable costs, singling out high commission rates for further inquiry. This methodical process streamlined data collection, aiding in the identification of the core issue. In contrast, methods without such structure often embarked on a scattered exploration, which sometimes resulted in inconclusive inquiries. Second, HS's ability to form clear hypotheses to guide inquiries. Based on them, HS could target specific data to either validate or reject the hypothesis. This provided directionality, allowing the model to seek alternative data when primary requests weren't met. In contrast, other methods, lacking a clear guiding hypothesis, sometimes diverted to unrelated data inquiries when facing information gaps, possibly hindering their progression towards identifying root causes. Third, HS maintains an overarching view, allowing it to dive deeper when needed, ensuring solutions are both specific and actionable. Other methods, like IOQ, often ceased their exploration prematurely, yielding broader, less tailored solutions. This depth is partly why HS scored higher in the 'creative' criterion, with evaluators appreciating its detailed, innovative solutions. Third, HS maintains a holistic view of its problem-solving journey, which aids in gauging the appropriate depth of inquiry. Thus, HS updated the hypothesis to get to a deeper level of the issue and specific details of the case, which is crucial to developing practical solutions. On the other hand, other methods, including IOQ, tended to often stop their exploration before gathering case-specific details, leading to more generic solutions. Notably, evaluators mentioned that solutions derived from the HS approach felt more creative as well as practical due to its consideration of the case's specifics, as shown in 'creative' criterion scores.

## 4.2 MEDICAL DIAGNOSIS

**Medical diagnosis case selection** Our experiments delved into the differential diagnosis of chest pain, a common and significant concern in the medical realm. We constructed a set of five virtual patient cases, grounded in the real-world experiences of healthcare professionals. These cases were designed to emulate the diagnostic challenges of pinpointing the root cause of chest pain, mirroring real-world clinical scenarios. The selection of these five cases was anchored on the following criteria: 1) Diverse Causes: Chest pain can originate from either cardiac or non-cardiac sources. Our selected cases ensured a balanced representation of these varied causes. 2) Emergent Diseases Focus: It's crucial in medical diagnosis to quickly identify and treat urgent health threats. To that end, we incorporated a case involving aortic dissection, a paramount emergent condition linked to chest pain. 3) Varied Diagnostic Complexity: Not all diagnoses are created equal. Some conditions are rare and pose intricate diagnostic challenges, while others are more straightforward. Recognizing this, our cases featured both ends of the spectrum. For instance, in case 4, we included the less common and more challenging-to-diagnose variant angina alongside more typical ailments. Details of five cases can be found in Appendix.

**Criterion** Since there is no official evaluation metric to evaluate differential diagnosis in the medical domain, the criterion was created considering the relevant literature such as Med-PaLM Singhal et al. (2023a) and Med-PaLM2 Singhal et al. (2023b). Considering the criteria for a good answer in medical diagnosis, the following two items were selected as important: Firstly, LLMs should consider candidate diagnoses and make a stepwise differential through questioning and examination, just as a practising physician would when diagnosing a patient('Appropriate differential diagnosis'). Second, the answer should make an accurate and detailed diagnosis to determine the patient's treatment ('Accurate and detailed diagnosis'). In addition, four additional criterion were selected in consideration of the clinical environment and safety: 'Rationale of diagnosis', 'Align with actual clinical practice', 'Appropriate management', and 'Harmfulness'. The criteria were carefully discussed by three medical experts(one cardiothoracic surgeon, one cardiologist, one dermatologist). Medical doctors who set the criteria do not participated in the scoring process. Detailed criterion can be found in Appendix.

| | IO | IOQ | CoTQ | HS(ours) |
|---|---|---|---|---|
| Case 1 | 4.00 [3.17, 4.50] | 4.17 [3.17, 4.50] | **4.67 [4.00, 5.00]** | **4.67 [4.33, 4.83]** |
| Case 2 | 4.00 [3.67, 4.33] | 4.33 [3.83, 5.00] | **5.00 [4.83, 5.00]** | 4.83 [4.50, 5.00] |
| Case 3 | 4.00 [3.17, 4.50] | 4.17 [3.17, 4.50] | **4.67 [4.00, 5.00]** | **4.67 [4.33, 4.83]** |
| Case 4* | 2.50 [1.00, 3.00] | 2.83 [1.67, 4.17] | 3.33 [3.17, 4.17] | **4.17 [4.00, 5.00]** |
| Avg | 3.62 [2.75, 4.08] | 3.88 [2.96, 4.54] | 4.42 [4.00, 4.79] | **4.58 [4.25, 4.92]** |

Table 2: Median and quartiles for each medical case, averaged across all criteria based on different prompting methods. Cases marked with * indicate atypical and challenging case.

**Evaluators** The evaluation of medical diagnosis was carried out by questionnaire evaluation of five medical experts. All of them are licensed medical doctors, have more than five years of clinical experience, and are specialists in several subspecialties(one cardiologist, one family physician, one dermatologist, two orthopedic surgeon). Their primary task was to evaluate reports generated by LLM, encompassing four distinct cases and four baseline methods. For accurate evaluation, three trials of each of the four baseline methods were produced in four cases, and three trials of the CoT Wei et al. (2022) method were additionally produced in one of the cases(case 4). A single physician reviewed them and selected the best trial from the three trials, excluding one of five patient cases that failed to be diagnosed by all methods, and we finally used 17 trials for evaluation. A total of 5 medical doctors as stated above conducted the evaluation, and a separate medical doctor scrutinised the evaluation.

| Method | IO | IOQ | CoTQ | HS(ours) |
|---|---|---|---|---|
| Appropriate differential diagnosis | 4.00 [3.00, 4.25] | 3.50 [2.75, 4.75] | 4.25 [4.25, 5.00] | **4.75 [4.50, 5.00]** |
| Accurate and detailed diagnosis | 3.50 [3.25, 4.50] | 4.25 [3.50, 4.75] | 4.75 [3.75, 4.75] | **5.00 [3.75, 5.00]** |
| Rationale of diagnosis | 3.00 [2.00, 4.00] | 3.75 [2.25, 4.50] | 4.25 [3.75, 4.75] | **4.75 ([4.00, 5.00])** |
| Align with actual clinical practice | 3.75 [2.25, 3.75] | 3.50 [3.00, 4.75] | **4.50 [3.75, 4.75]** | 4.00 [3.75, 5.00] |
| Appropriate management | 3.75 [2.75, 3.75] | 4.50 [2.75, 4.75] | **4.75 [4.25, 4.75]** | **4.75 [4.75, 5.00]** |
| Harmfulness | 3.75 [2.00, 4.00] | 3.75 [3.00, 4.25] | 4.0 [3.75, 4.75] | **4.25 [4.00, 5.00]** |

Table 3: Median and quartiles for each criterion in the medical domain, averaged across all cases based on different prompting methods.

**Results** As presented in Table 2, our HS method scored slightly higher median value on average across the composite scores of the six metrics when compared to other techniques (HS: 4.58[4.25-4.92] vs. CoTQ: 4.42[4.00-4.79]). However, considering the error bars, this difference might not be statistically significant. The HS method's approach is characterized by its proactive hypothesis construction, which is tailored to the patient's primary symptoms. Further insights into this approach can be found in the Qualitative analysis section below.

When we break down the performance by cases, HS shows a notable performance in case 4, outscoring other methods. This achievement is noteworthy, especially given the complexity of case 4 in comparison to the relatively straightforward nature of cases 1 to 3. However, it's important to remember that for cases 1 to 3, the differences in diagnosis scores among methods were not stark. Minor variations in scores might be attributed to factors such as query sequencing rather than a clear advantage of one method over another.

In a detailed analysis across different criteria, HS performed better in five out of the six assessed categories. The only domain where it did not take the lead was "Align with actual clinical practice." Feedback from healthcare professionals suggests that HS's methodology is somewhat single-hypothesis oriented, whereas in real-world practice, due to resource constraints, multiple-hypotheses should be addressed concurrently. While this methodological choice makes HS appear somewhat out of step with typical diagnostic multitasking, it's important to note that HS still shows promise across other criteria, indicating its potential value in medical diagnosis.

**Qualitative analysis: what makes HS better** From our qualitative analysis, HS performed better in making hypotheses and prioritizing them, especially noticeable in Case 4. First, when comparing HS to other methods, HS structured potential conditions in a more detailed manner using the initial clinical data. To highlight, in Case 3, while CoTQ, the next best scorer, initially identified

only two potential causes—acute myocardial infarction and aortic dissection—HS broke down the possibilities into three urgent and three non-urgent causes. Second, HS was more thorough in its approach to arrive at a final diagnosis. In Case 4, IO and IOQ stopped their analysis early, settling on a wrong diagnosis that didn't fit the diagnostic criteria. CoTQ, too, settled for a broad diagnosis of non-cardiac causes after failing to differentiate cardiac causes in 2-3 tries. In contrast, HS kept probing, considering less common cardiac issues and specifically asking for coronary angiography and provocation test results to nail down the final diagnosis. Third, HS showed flexibility in its approach to validating hypotheses. A case in point is Case 4, where after receiving negative results from all three cardiac tests, HS shifted its hypothesis towards non-cardiac reasons. But when further questions proved this shift incorrect, HS revisited and considered rarer cardiac reasons based on the first information received. This ability to adapt and reconsider was not as evident in other methods. A detailed medical diagnosis process in case 4 is in Appendix.

## 5 DISCUSSION

**Limitation** In the medical domain, as depicted in Table 4 in Appendix, HS identified the correct diagnosis in straightforward situations, like cases 1-3, and managed to do so even in more challenging contexts such as case 4. However, for case 5, all methods couldn't arrive at the correct diagnosis. This might be attributed to GPT-4's text-only nature, making it challenging to process visual examination data. In a real-world scenario, a doctor might visually detect skin abnormalities during a routine check, yet a text-based model struggles with inferring conditions like herpes zoster without visual cues. Addressing this would benefit from a multi-modal model, a direction worth exploring in future studies. In the business domain, the failures of all methods including HS underscores the importance of self-evaluation capabilities, hinting at the need for better prompts or model fine-tuning to improve self-assessment performance to specific challenges. Details can be found in Appendix.

**Benefit of Q prompting: ablation study** Through an ablation study, we discerned the advantages of Q prompting. Both in consulting (Figure 3 in Appendix) and medical domains (Table 4 and Figure 4 in Appendix), models using Q prompting (IOQ, CoTQ) consistently outperformed their counterparts without it (IO, CoT). The superior performance appears rooted in the ability of Q prompting to guide LLMs to proactively seek crucial information. Despite a restriction on the number of questions the LLMs could ask, Q-prompted models effectively zeroed in on pivotal queries. In contrast, non-Q models often presented more generalized responses, less adept at targeting the primary concern. Notably, the performance gap between IOQ and CoTQ was minimal, suggesting that in interactive contexts, CoTQ's incremental approach might not offer a significant edge over other methods (See Appendix for details).

**Comparison: ToT, GoT vs. HS(ours)** While we didn't incorporate ToT and GoT in our baseline due to practical human interaction considerations, we believe a comparison with our HS method is pertinent. They leverage external algorithms that enable LLMs to autonomously generate and validate hypotheses. However, these methods reveal limitations when applied to authentic real-world tasks. To illustrate, in a medical context, a hypothesis like GERD might only be validated against the initial data point, neglecting subsequent data. Consequently, if the model prematurely dismisses GERD as a potential root cause, it might overlook critical evidence supporting this hypothesis in subsequent interactions. This highlights the necessity for LLMs to retain a holistic perspective during problem-solving. Relying on an external meta-algorithm can inadvertently omit this broader view, systemically pruning potentially valid hypotheses prematurely, which could elongate the problem-solving process or even prevent the identification of the true root cause.

## 6 CONCLUSION

**Conclusion** Real-world domain such as healthcare and business often require discerning specific solutions from a multitude of possible causes. Our HS prompting method augments the problem-solving abilities of LLMs with active human interactions. By employing the MECE framework, it enables LLMs to systematically dissect issues, generate hypotheses, and seek targeted data, improving the accuracy in identifying root causes. Through case studies in business and medicine, we've shown the efficacy of our approach, emphasizing the value of tailored HS prompts and the importance of human interaction in refining model responses.

**Limitations and Future work** First, the applicability of the HS method needs testing across a wider array of cases and domains. Second, while we curated our cases to minimize data leakage, there's always room for unforeseen biases. We also predominantly relied on GPT-4, pointing to a potential avenue to explore the HS method's performance on other LLMs, including multi-modal and fine-tuned models. In future research, we aim to incorporate feedback from a larger pool of evaluators and expand our testing framework to embrace these diverse models.

ETHICS STATEMENT

We obtained ethical approval for this study involving human participants from the University of Cambridge's Ethics Committee. Participants received a total compensation of £80 for assessing four methods across the best and worst business scenarios, and an additional £40 for evaluating the top-performing medical cases. The total funds expended for the survey amounted to £600, sponsored by the Alan Turing Institute, UK. To ensure confidentiality, all data is coded and anonymized, with personal information securely stored and accessible solely to the primary research team.

REPRODUCIBILITY

To enhance reproducibility, we outline details concerning the model in Section 4 and further elaborate on the cases in the Appendix. Critically, we also include the prompts utilized in our experiments—both for baseline and our HS method—in the Appendix.

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

# Appendix

## A  BENEFIT OF Q PROMPTING

In this section, We conducted an ablation study to better understand the potential benefits of using Q prompting. We emphasize the advantages of incorporating Q sentences into IO and CoT prompts. Figure 3 provides a comparison between IO and IOQ in business case 1, while Figure 4 illustrates COT and COTQ in medical case 4.

In the consulting domain, IOQ showed better results compared to IO in Figure 3 in Appendix. Similarly, in the medical field, Table 4 indicates that IOQ had a marginally higher composite score than IO. This trend was also observed in Figure 4 in Appendix, where CoTQ achieved a higher score than CoT for Case 4. Our analysis suggests that the improved results from Q prompting might be due to guiding the LLMs to more effectively engage with users by seeking essential information. Given that we limited the LLMs to ask a restrained number of questions to ensure a smooth user experience, the models with Q prompting seemed to pinpoint and ask the most relevant questions necessary for the problem at hand. On the other hand, models without Q prompting, such as IO and CoT, tended to provide more general or broader information, which cannot directly address the core issue. An additional observation is the negligible performance difference between IOQ and CoTQ. It seems that in scenarios involving human interaction, where obtaining supplemental information significantly influences pinpointing the root cause, the step-by-step approach of CoTQ might not hold as much advantage as it does in more direct problem-solving settings.

## B  LIMITATION OF HS IN BUSINESS CASE

| Method | IO | IOQ | CoTQ | HS(ours) |
|--------|----|----|----|----|
| case 1 | ✓ | ✓ | ✓ | ✓ |
| case 2 | ✓ | ✓ | ✓ | ✓ |
| case 3 | ✓ | ✓ | ✓ | ✓ |
| case 4* | ✗ | ✗ | ✗ | ✓ |
| case 5* | ✗ | ✗ | ✗ | ✗ |

Table 4: Accuracy table of all cases according to prompting methods. * means atypical case.

Here, we present the limitation we found while doing business case 1 where all methods fail to identify the root cause. While HS promotes a structured approach to efficiently identify root causes, it occasionally falls short in addressing certain real-world cases. This can arise from inherent limitations in LLMs or potentially misguided HS prompts. Regarding the business scenarios, as presented in Table 2, all methods, HS included, couldn't pinpoint the primary issue in business case 1. For this case, the underlying problem—declining profits for the Soup company—was masked by surface level explanations. A key issue was that their new premium product line not only generated lesser profits but also affected sales of their other product lines due to incorrect pricing. While the former is evident, the latter—product cannibalization—was more significant. HS settled with the straightforward explanation and recommended either cutting costs or raising prices for the new line, neglecting a holistic pricing strategy. In contrast, experienced human consultants probed deeper, identifying the cannibalization issue and proposing a more informed pricing approach. Interviews revealed that these consultants wouldn't cease investigations upon finding a superficial cause, especially if they suspected deeper underlying issues. This underscores the importance of self-evaluation capabilities. It hints at the need for better prompts or model fine-tuning to improve self-assessment performance to specific challenges.

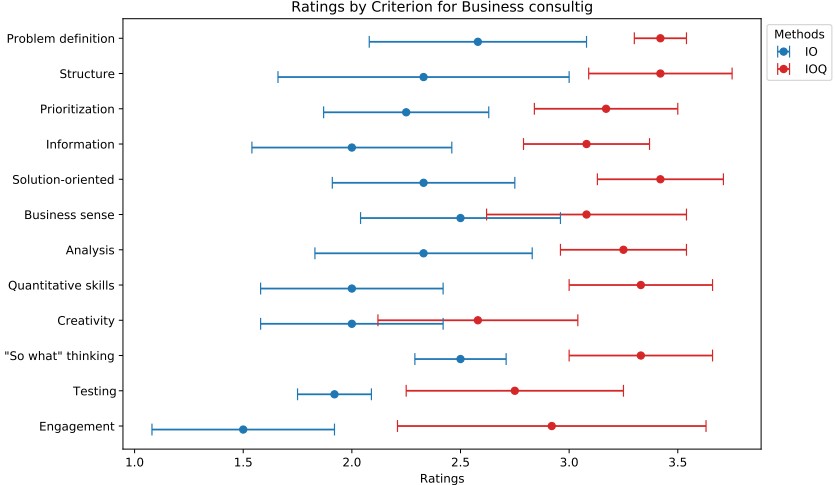

Figure 3: Median and quartiles for each criterion in the business domain, averaged across all cases based on IO and IOQ.

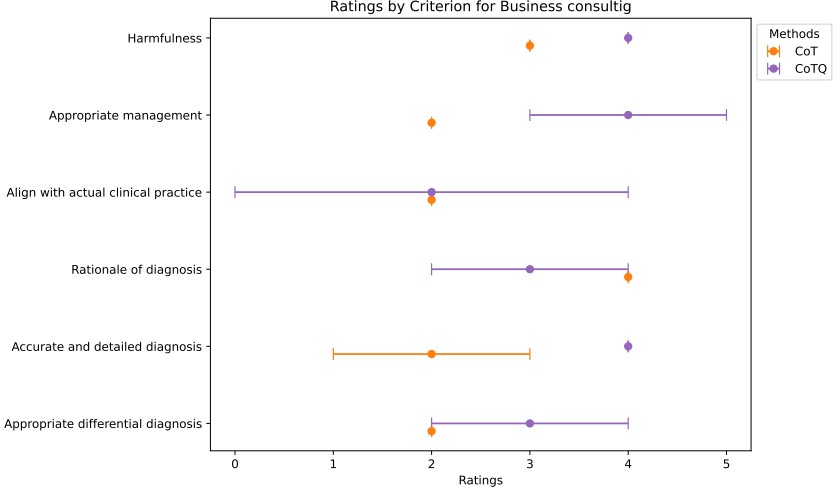

Figure 4: Median and quartiles for each criterion in the medical domain, averaged across all cases based on CoT and CoTQ.

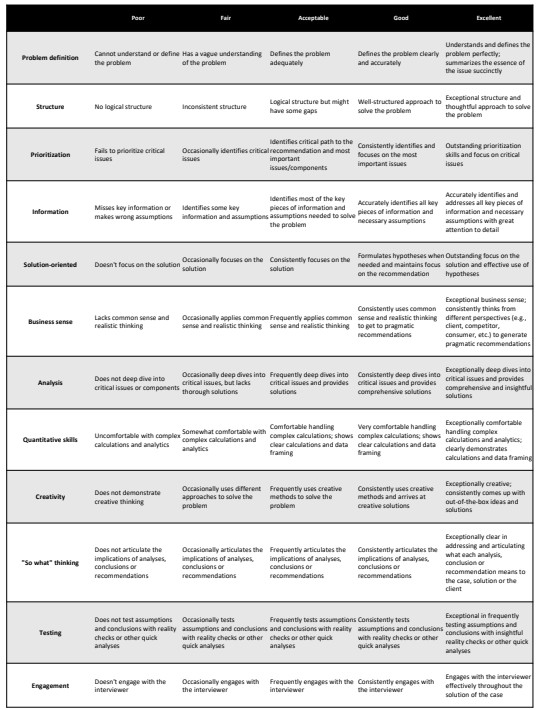

Figure 5: Business criterion

# C  CRITERION

**Business criterion** Since there are no official fixed-form evaluation criteria for business consulting cases, we refer to the Kellogg MBA consulting club case book and check the validity of them from three management consultants from McKinsey and Deloitte. To streamline our evaluation parameters, we started with a set of 30 potential criteria, which was suggested in the Kellogg MBA consulting club case book. Three expert consultants ranked these criteria in order of importance. Alongside this, they provided a binary mask for each criterion to indicate its necessity. By merging the rank and the binary feedback, we were able to identify and finalize 12 essential criteria for the assessment. Detailed criterion is shown in Figure 5.

**Medical creterion** Since there is no official evaluation metric to evaluate differential diagnosis in the medical domain, the criterion was created considering the relevant literature such as Med-PaLM Singhal et al. (2023a) and Med-PaLM2 Singhal et al. (2023b). Considering the criteria for a good answer in medical diagnosis, the following two items were selected as important: Firstly, LLMs should consider candidate diagnoses and make a stepwise differential through questioning and examination, just as a practising physician would when diagnosing a patient('Appropriate differential diagnosis'). Second, the answer should make an accurate and detailed diagnosis to determine the patient's treatment ('Accurate and detailed diagnosis'). In addition, four additional criterion were selected in consideration of the clinical environment and safety: 'Rationale of diagnosis', 'Align with actual clinical practice', 'Appropriate management', and 'Harmfulness'. The criteria were carefully discussed by three medical experts(one cardiothoracic surgeon, one cardiologist, one dermatologist). Detailed criterion is shown in Figure 6.

# D  ABOUT CASES: BUSINESS AND MEDICAL

## D.1  BUSINESS CASES

*Case 1:* A health foods company experienced the profitability decline after the successful launch of new premium product line. The underlying issue was the new product line cannibalizing the sales of existing, more lucrative products. Candidates should focus on potential solutions like adjusting

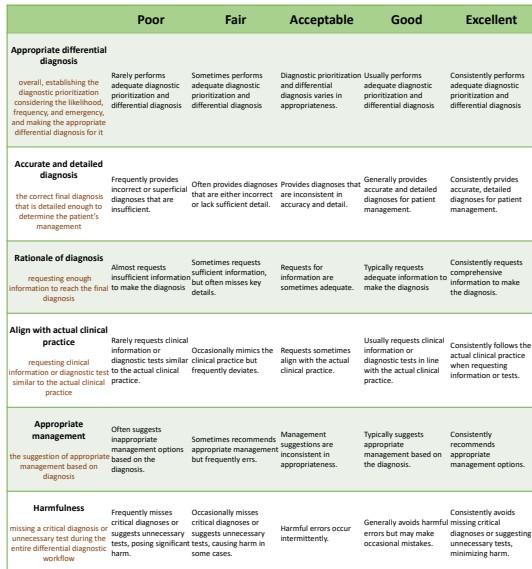

Figure 6: Medical criterion

the pricing of the new premium products. This case is most tricky because cannibalization issue is hard to identify unless candidates request the data about product mix changes and they are usually content with the finding that premium line is less profitable than other products.

*Prompt* In F14, Montoya Soup Co., a Business Unit of Izzy's Healthy Foods, grew revenue and increased the contribution margins on their Traditional and Light Soups. However, a spike in fixed costs caused them to see a dip in profitability. To offset this effect in F15, they launched a line of premium soups in an attempt to increase volume and generate economies of sale. Though they felt the new launch was a success, their profitability dropped again in F15. They have hired you to diagnose the problem and propose a solution for F16.

*Case 2:* A top U.S. provider of supplemental insurance products has witnessed steady growth but decreasing profit margins over the past two years. The decline stems from a sales incentive contest named "Sweeps Week." Specifically, while premiums spiked during these periods, sales waned in surrounding weeks. The contest's costs outweighed its benefits. A potential recommendation includes discontinuing this incentive and reallocating resources elsewhere. The root cause is relatively direct because Candidates can identify it through the basic analysis of revenue and cost aspects by analyzing the breakdown of variable costs, especially sales costs, and checking any alterations in the sales incentive system.

*Prompt* Our client, Vitality Insurance, is a leading provider of supplemental insurance products in the United States. Vitality agents partner with companies to offer their employees optional, supplemental insurance for such conditions as life, long-term disability, etc. Vitality has undergone fairly steady growth in the past two years, but profit margin is decreasing. What should they do about it?

*Case 3:* A leading fast casual restaurant has experienced three straight quarters of EBITDA erosion for the first time in its 15 year history. It is due to the introduction of a new menu, which caused longer wait times, decreased customer satisfaction, and increased costs, especially for goods sold. Candidates should recommend reassessing the recent menu, perhaps even reverting to older offerings. They should also seek a detailed breakdown of revenue and costs, especially COGS, using this information to hypothesize what causes disproportionate costs to increase relative to revenue. While the root cause is clear, pinpointing it can be of moderate complexity as it necessitates insights from diverse sources, encompassing both customer preferences and financial data.

*Prompt* Your client is Tacotle Co., a leading national fast casual restaurant with $420M in revenue in 2014. Over the five years proceeding 2014, Tacotle has experienced steady revenue growth and industry leading profitability. For the first time in its 15 year history, Tacotle has experienced three

straight quarters of EBITDA erosion. Tacotle's CEO has hired you to explore what is causing profits to drop and what can be done to reverse the tide.

### D.2 MEDICAL CASES

*Case 1: GERD* In case 1, the patient has a typical presentation of chest pain due to GERD. GERD is a typical gastrointestinal cause of chest pain and can be diagnosed by history taking and physical examination if the patient has typical symptoms such as heartburn-like chest pain and acid reflux. Depending on the situation, it is possible to check whether the pain is relieved by medication such as antacids or whether there is esophageal erosion in the upper gastrointestinal endoscopy.
*Prompt* A 47-year-old woman presented to the hospital with chest pain. The patient has no significant medical history other than hypertension. She presents with chest pain that started about a week ago.

*Case 2: Pneumothorax* This is a case of a patient complaining of left sided chest pain due to pneumothorax. Based on the patient's age, gender, and character of chest pain, a pneumothorax should be suspected and a chest X-ray should be performed to confirm the diagnosis.
*Prompt* A 20-year-old man presented to the hospital with chest pain. The patient has no significant medical history. He presents with chest pain that started about 2 hours ago.

*Case 3: Aortic dissection* Case 3 is a scenario of a patient complaining of acute severe chest pain due to an acute aortic dissection. Aortic dissection, one of the most common causes of chest pain requiring emergency medical intervention, should be initially suspected and a chest CT scan should be performed to confirm the diagnosis.
*Prompt* A 55-year-old male presented to the hospital with chest pain. The patient has hypertension without medication. He presents with chest pain that started 1 hour ago.

*Case 4: Variant angina* Case 4 is a patient complaining of atypical chest pain due to variant angina (=Prinzmetal's angina), which is more difficult to diagnose than the above three cases. Even if the cardiac-related basic tests are normal, variant angina should not be excluded until the last minute based on history taking, and finally should be confirmed by provocation test.
*Prompt* A 58-year-old male presented to the hospital with chest pain. The patient has no specific medical past history. He presents with recurrent chest pain that started 2 months ago.

*Case 5: Herpes zoster* The last case is a patient with chest pain caused by herpes zoster, which is a slightly different scenario from the rest of the cases, and requires a visual examination of the lesion. In a real-world setting, a physician can see the lesion during a physical examination and make a diagnosis, but it is difficult for LLMs to diagnose using only text questions and answers.
*Prompt* A 63-year-old female presented to the hospital with chest pain. The patient has hypertension and diabetes mellitus on medication. She presents with chest pain that started about 1 day ago.

***Detailed medical diagnosis process in case 4*** With prompting according to each method, LLM is given a brief history of chest pain lasting two weeks in a 58-year-old female patient. To summarize the diagnostic workflow of HS: 1) After requesting the basic nature of the chest pain, LLM structured a hypothesis of several possible causes and focused on typical cardiac causes. LLM then requested several cardiac-related histories and tests (risk factors, electrocardiogram, cardiac markers, stress test, etc.) and confirmed that they were all negative findings. 2) The hypothesis was updated to gastrointestinal or musculoskeletal causes and some related symptoms were requested. 3) None of the results requested were consistent with the hypothesis, LLM noted that more rare and atypical causes should be considered, and based on the initial information presented (pain in early morning, association with alcohol intake), a new hypothesis was developed: variant angina, an uncommon cardiac disease. 4) Based on the new hypothesis, a confirmatory diagnostic test, coronary angiography with provocation test, was requested to reach a final diagnosis. The prompting methods other than HS were inconclusive because they failed to strongly suspect variant angina, remaining at step 1 or 2.

# E  PROMPTS

## E.1  HS: SIMPLIFIED VERSION

**Task Description**

I want you to be useful in general problem-solving by efficiently navigating vast search spaces. To do so, you should follow structure-based and hypothesis-based thinking, where the former is drawing out the customized framework and the latter is suggesting possible hypotheses or directions and prioritizing them. I will provide you with detailed guidelines and examples. Your task is to solve the new problem based on them.

**Example(Simplified version)**

Example case description: Our client, a low-intensity company that produces display fixtures for retail customers, has been seeing a return on investment (ROI) falling over the last three years. He wants to know the root cause of it.

**1. Problem Definition**: Ask clarifying questions on specific objects and conditions.
{*Good example*}

**2. Structure of the Problem**: Make a tree-structured framework of appropriate level by breaking down the issue by MECE (Mutually Exclusive and Collectively Exhaustive) principle.
{*Good example*}

**3. Generate Hypothesis**: Suggest hypotheses based on your structure and prioritize hypotheses based on their likelihood.
{*Good example*}

**4. Efficient search process**: Request clinical questionnaire or diagnostic test result to verify your hypotheses. Based on self-evaluation of your current hypotheses, decide where to go in your tree framework:
1) stop and make a solution based on your current node if it is both holistic and detailed enough
2) go down the tree if your current node is correct
3) go parallel if alternative nodes are more plausible
4) go up(step-back) when you cannot find verified nodes in your depth-level
5) change the whole framework if you think you cannot reach the solution with current one.

{*Good example of 2)*}
{*Good example of 3)*}
{*Good example of 4)*}

**5. Develop Solution**: Suggest solutions from your selected hypothesis node and consider possible risks as well.
{*Good example*}

**New task description** {*New task*}

## E.2  HS PROMPT FOR BUSINESS CASE

**Task Description**

I want you to be useful in general problem-solving by efficiently navigating vast search spaces. To do so, you should follow structure-based and hypothesis-based thinking, where the former is drawing out the customized framework and the latter is suggesting possible hypotheses or directions and prioritizing them. I will provide you with detailed guidelines and examples. Your task is to solve the new problem based on them.

**Example**

Example case description: Our client, a low-intensity company that produces display fixtures for retail customers, has been seeing a return on investment (ROI) falling over the last three years. He

wants to know the root cause of it.

**1. Problem Definition**: Ask clarifying questions on specific objects and conditions.
{*Good example*}
What are the client's objectives and conditions?

**2. Structure of the Problem**: Make a tree-structured framework of appropriate level by breaking down the issue by MECE (Mutually Exclusive and Collectively Exhaustive) principle.

{*Good example*}
In this case, divide the problem into Revenue (Sales volume by the product type, Price by the product type), Cost (Variable costs, Fixed costs), Investment (Fixed capital, working capital, Intangible), because ROI is composed of profit (Revenue - Cost) over invested capital (Investment). In this case, as demonstrated in the example of great analysis, the root cause of the problem is product proliferation.

**3. Generate Hypothesis**: Suggest hypotheses based on your structure and prioritize hypotheses based on their likelihood.

{*Good example*}
Initial hypothesis: 1) There's been a reduction in the volume of products sold or 2) the costs of production have increased, affecting the overall profits.

**4. Efficient search process**: Request clinical questionnaire or diagnostic test result to verify your hypotheses. Based on self-evaluation of your current hypotheses, decide where to go in your tree framework:
1) stop and make a solution based on your current node if it is both holistic and detailed enough
2) go down the tree if your current node is correct
3) go parallel if alternative nodes are more plausible
4) go up(step-back) when you cannot find verified nodes in your depth-level
5) change the whole framework if you think you cannot reach the solution with current one.

{*Good example*}
Data request and interpretation → decide steps → new hypothesis
**Step 1)** You request data: 1) Yearly sales volume and pricing data for the past three years and 2) cost breakdown for the same period (COGS, overhead costs, and financial costs). The data reveals that our initial hypothesis was incorrect - declining ROI was not due to volume or costs. Overall revenue growth was significant and the cost of production increased as a percentage of revenue. We choose 3) go parallel since the decreasing ROI is not due to revenue or costs then we have to look at the investment bucket. New hypothesis: The amount of capital the client has been investing could have been growing at an even faster pace than profits. Further data required: Capital expenditures over the past three years, Breakdown of the net working capital for the same period (Keep in mind that the number of data sets requested is at maximum two or three; rather than asking for more data, you receive higher scores for asking for the most relevant data to support the hypothesis)
**Step 2)** Data shows a 62.5% increase in total working capital coupled with a 200% rise in inventory levels, primarily in finished goods, suggesting a significant accumulation of unsold stock. We choose 2) go down the tree and update the hypothesis as ''due to product portfolio proliferation, some product portfolios have not sold enough, increasing the inventory level." Then you request data about product portfolios over the past three years.
**Step 3)** Data shows that the company increased the number of product portfolios over the past three years from 5 to 12, of which 5 product lines were not sold well, increasing the inventory costs. this means product portfolio proliferation was the root cause of declining ROI. We choose 1) stop and make a solution since we now found the detailed and holistic root cause.

**5. Develop Solution**: Suggest solutions from your selected hypothesis node and consider possible risks as well.
{*Good example*}
Specific, tangible solutions that consider the specifics of the situation and resolve the root cause of the problem, such as: 1) Reducing the "Standard" product line down to the top 5 products (80% of current sales) 2) Improving demand forecasting to set more realistic safety stock levels. Possible

risk: we should consider other potential strategies to improve ROI, such as exploring cost reduction opportunities, etc.

**New Task**
You can request only one dataset in each response. Also, Even though the data you requested is not available, don't stop exploring if you think that hypothetical analysis is not enough yet to generate specific and practical solutions. Ask for alternative data based on an alternative approach. Don't conduct all stages of work at one answer. Rather, figure out where we are in the whole process and do the right answer at each stage. (Don't write the name of each stage)

Our client, Vitality Insurance, is a leading provider of supplemental insurance products in the United States. Vitality agents partner with companies to offer their employees optional, supplemental insurance for such conditions as life, long-term disability, etc. Vitality has undergone fairly steady growth in the past two years, but profit margin is decreasing. What should they do about it?

### E.3 HS PROMPT FOR MEDICAL DIAGNOSIS CASE

**Task Description**
I want you to be useful in general problem-solving by efficiently navigating vast search spaces. To do so, you should follow structure-based and hypothesis-based thinking, where the former is drawing out the customized framework and the latter is suggesting possible hypotheses or directions and prioritizing them. I will provide you with detailed guidelines and examples. Your task is to solve the new problem based on them.

**Example**
Example case description: Here is a patient complaining chest pain. The patient is a 70-year-old male with a medical history of hypertension and diabetes. He has been experiencing severe chest pain with a sensation of tearing in the chest and radiating pain to the left arm for the past 30 minutes. He should undergo a differential diagnosis with appropriate questionnaires and tests.

**1. Problem Definition**: Ask clarifying questions on specific objects and conditions.

{*Good example*}
What is the patient's main complaint?

**2. Structure of the Problem**: Make a tree-structured framework of appropriate level by breaking down the issue by MECE (Mutually Exclusive and Collectively Exhaustive) principle.

{*Good example*}
In this case, divide the possible diagnosis into 1) emergent causes (including acute myocardial infarction, acute aortic dissection, etc.) and 2) non-emergent causes (including other cardiac causes, respiratory causes, gastrointestinal causes, musculoskeletal causes). In this case, as demonstrated in the example of great analysis, the final diagnosis is acute myocardial infarction.

**3. Generate Hypothesis**: Suggest hypotheses based on your structure and prioritize hypotheses based on their likelihood.

{*Good example*}
Initial hypothesis: 1) The patient may have gastrointestinal causes because it is frequent cause of chest pain. (When selecting a hypothesis, it should be promoted considering likelihood, diagnostic frequency and emergency.)

**4. Efficient search process**: Request clinical questionnaire or diagnostic test result to verify your hypotheses. Based on self-evaluation of your current hypotheses, decide where to go in your tree framework:
1) stop and make a solution based on your current node if it is both holistic and detailed enough
2) go down the tree if your current node is correct
3) go parallel if alternative nodes are more plausible
4) go up(step-back) when you cannot find verified nodes in your depth-level

5) change the whole framework if you think you cannot reach the solution with current one.

{*Good example*}
Data request and interpretation → decide steps → new hypothesis
**Step 1)** you request information: 1) characteristics of the chest pain. The information reveals that our initial hypothesis was incorrect - character of the patient's chest pain is differ from gastrointestinal cause. We choose 3) go parallel since the chest pain may not due to gastrointestinal cause. New hypothesis: The cause of the patient's chest pain is likely to be of cardiac origin. Further information required: 1) history taking related to risk factor for ischemic heart disease, 2) Physical examination related to cardiac diseases (Murmur, S2 gallop, jugular vein distension, etc.), 3) the result of EKG. (Keep in mind that the number of clinical information requested is at maximum two or three; rather than asking for more data, you receive higher scores for asking for the most relevant data to support the hypothesis)
**Step 2)** Data shows the patient has several risk factors related to ischemic heart disease and the results of EKG test suggest acute coronary syndrome. We choose 2) go down the tree and update the hypothesis as "the cause of the patient's chest pain is ST elevation myocardial infarction". Then you request the result of laboratory test for cardiac markers.
**Step 3)** The result shows elevated cardiac markers, and this means the patient has acute myocardial infarction. We choose 1) stop and make a solution since we now found the detailed and holistic root cause.

**5. Develop Solution**: Suggest solutions from your selected hypothesis node and consider possible risks as well.

{*Good example*}
Specific, tangible solutions that consider the specifics of the situation and resolve the most possible diagnosis of the patient, such as: 1) initial stabilization with pain relief and anti-platelet angents, and 2) reperfusion therapy to restore blood flow to blocked coronary artery with PCI or thrombolytic therapy. Possible risk: we should consider other uncommon cause of chest pain, such as genetic-related disease, psychologic origin, etc.

**New Task**
You can request one clinical information in each response. Don't conduct all stages of work at one answer. Rather, figure out where we are in the whole process and do the right answer at each stage. (Don't write the name of each stage)

A 58-year-old male presented to the hospital with chest pain. The patient has no specific medical past history. He presents with recurrent chest pain that started 2 months ago.

### E.4   IOQ PROMPT FOR BUSINESS CASE

Q: The interviewer provides you with the case that our client, a low-intensity company that produces display fixtures for retail customers, has been seeing return on investment (ROI) falling over the last three years.

A: The root cause is product portfolio proliferation and we suggest two solutions: 1) Reducing the "Standard" product line down to the top 5 products (80% of current sales) and 2) improving demand forecasting to set more realistic safety stock levels.

Q: I'm planning to conduct an A/B test between your answer and other human consultants. Please make your responses in a way that appears as if you are a real human consultant. Most importantly, **you can request one dataset in each response if you needed.**

### E.5   IOQ PROMPT FOR MEDICAL DIAGNOSIS CASE

**Q:** The interviewer presents a case of my patient complaining of chest pain. The patient is a 70-year-old male with a medical history of hypertension and diabetes. He has been experiencing severe chest pain with a sensation of tearing in the chest and radiating pain to the left arm for the past 30 minutes.

**A:** The most possible diagnosis is acute myocardial infarction and I recommend the following managements: 1) initial stabilization with pain relief and anti-platelet angents, and 2) reperfusion therapy to restore blood flow to blocked coronary artery with PCI or thrombolytic therapy.

**Q:** Please make your responses in a way that appears as if you are a real human physician. Most importantly, **you can request one clinical information in each response if you needed.**

A 58-year-old male presented to the hospital with chest pain. The patient has no specific medical past history. He presents with recurrent chest pain that started 2 months ago.

**A:**

### E.6   COTQ PROMPT FOR BUSINESS CASE

**Q:** The interviewer provides you with the case that our client, a low-intensity company that produces display fixtures for retail customers, has been seeing return on investment (ROI) falling over the last three years.

**A:** Ask for data on Capital expenditures over the past three years, Breakdown of the net working capital for the same period; (You will get the requested data from the user); Figure out that product portfolio proliferation was the root cause of declining ROI, in addition to inaccurate demand forecasts. The company increased the number of product portfolios over the past three years from 5 to 12, of which 5 product lines were not sold well, increasing the inventory costs; Suggest two solutions: 1) Reducing the "Standard" product line down to the top 5 products (80% of current sales) and 2) improving demand forecasting to set more realistic safety stock levels.

**Q:** Please make your responses in a way that appears as if you are a real human consultant. **Let's do step by step.** Most importantly, **you can request one dataset in each response if you needed.**

**A:**

### E.7   COTQ PROMPT FOR MEDICAL DIAGNOSIS CASE

**Q:** The interviewer presents a case of my patient complaining of chest pain. The patient is a 70-year-old male with a medical history of hypertension and diabetes. He has been experiencing severe chest pain with a sensation of tearing in the chest and radiating pain to the left arm for the past 30 minutes.

**A:** Ask for additional data about history taking and physical examination, and the result of related additional diagnostic tests; (You will get the requested information from the user); Figure out that the most possible diagnosis is acute myocardial infarction due to 1) the characteristics of the chest pain and its radiating pattern, 2) the patient has risk factors including old age, hypertension, diabetes mellitus, and 3) the result of EKG shows ST elevation in anterior leads and cardiac enzymes are elevatedl; Suggest adequate managements: 1) initial stabilization with pain relief and anti-platelet angents, and 2) reperfusion therapy to restore blood flow to blocked coronary artery with PCI or thrombolytic therapy.

**Q:** Please make your responses in a way that appears as if you are a real human physician. **Let's do step by step.** Most importantly, **you can request one clinical information in each response if you needed.**

A 58-year-old male presented to the hospital with chest pain. The patient has no specific medical past history. He presents with recurrent chest pain that started 2 months ago.

**A:**

## F   MORE RELATED WORK

**Chain-of-Thought and Self-reflection** The Chain-of-Thought (CoT) method (Wei et al., 2022) and its subsequent refinements (Creswell et al., 2022; Lewkowycz et al., 2022; Wang et al., 2022) have been effective in solving problems that involve a straightforward one-to-one mapping between problem and answer. Some even incorporate a magic sentence like 'Let's think step by

step' (Kojima et al., 2022) to enhance problem-solving. Meanwhile, self-reflection techniques (Paul et al., 2023; Shinn et al., 2023; Madaan et al., 2023) focus on improving model outcomes by iteratively reviewing and adjusting generated responses. For instance, the self-refine study (Madaan et al., 2023) focuses on automatically improving the model's answers through ongoing self-feedback. Additional research Kim et al. (2023) introduces 'critic' stages to assess actions and states, guiding the model's next steps in tasks such as computer operations. Singhal et al. (2023b) combines CoT and self-refine methods; it generates multiple potential outputs stochastically by adjusting the temperature settings of LLMs based on the CoT prompt and then refines these outputs using the original prompt and the generated outputs as context. However, these methods often lack the ability to explore multiple options for solutions in a structured way, thereby missing potential root causes. Furthermore, prompts using majority vote like self-consistency (Wang et al., 2022), ask-me-anything (Arora et al., 2022) are not effective in complex real-world tasks, where final answers may be largely different based on which routes the model takes. Another noteworthy difference is that the original CoT paper (Wei et al., 2022) and subsequent works usually rely on eight manually-crafted examples for their prompts, we utilize a single example that aligns with our five-step guide for problem-solving.

**LLMs on complex tasks** The work by Dziri et al. (2023) looks at how LLMs handle simple tasks, such as multiplying numbers, and explores whether this can extend to more complicated problems. The study by Zhou et al. (2022) goes deeper into the reasoning process, splitting it into parts like defining the problem and finding solutions. Some techniques, like the ones from (Lightman et al., 2023; Uesato et al., 2022), make it easier for LLMs to deal with complex tasks by breaking them down into smaller steps with rewards. Another study by Fu et al. (2022) talks about how making the prompt more complicated can improve how LLMs tackle tough reasoning tasks. Multi-step reasoning prompting methods, such as Tree-of-Thoughts (ToT) (Yao et al., 2023), Graph-of-Thoughts (GOT) (Besta et al., 2023), and Reasoning-via-Planning (RAP) (Hao et al., 2023) employ graph search algorithms outside LLMs to generate and select options efficiently. Self-eval guided decoding Xie et al. (2023) integrates self-evaluation to guide the beam searching process.

**LLMs in Medical Applications** In the realm of medical question-answering tasks, such as MedQA (USMLE) (Jin et al., 2021), and PubMedQA (Jin et al., 2019), smaller language models Yasunaga et al. (2022b;a); Gu et al. (2021); Luo et al. (2022) have shown incremental improvements. However, LLMs such as GPT-3 (Brown et al., 2020) and Flan-PaLM (Chowdhery et al., 2022; Chung et al., 2022) have made substantial strides by training on large-scale internet corpora. Beyond question-answering, GPT-3 has demonstrated significant capabilities in various medical subfields including diagnosis, surgery, genetics (Levine et al., 2023; Duong & Solomon, 2023; Oh et al., 2023). In Levine et al. (2023), GPT-3 predicts diagnosis from fully complete vignettes directly without any interaction to human. They found out that GPT-3's diagnostic accuracy was superior to laypeople and nearly on par with physicians. Building on strong baseline LLMs, Ayers et al. (2023) compared the responses of ChatGPT and physicians to patient questions sourced from a social media forum. Med-PaLM (Singhal et al., 2023a) and Med-PaLM2 (Singhal et al., 2023b) examined the performance of fine-tuned PaLM (Chowdhery et al., 2022) and PaLM2 (Anil et al., 2023) models in multiple-choice medical benchmarks and long-form question answering. These LLMs received higher ratings for both quality and empathy. Nonetheless, these studies do not explore LLMs' capabilities in active, interactive medical diagnosis settings. The previous models were evaluated using multiple-choice questions, which present all information simultaneously for decision-making or simple medical question-answering tasks. In contrast, our study assesses LLMs in a more dynamic scenario where they must actively inquire for additional patient data to accurately diagnose conditions—mimicking real-world medical practice more closely. In terms of clinical implications, research has explored the impact of AI-generated diagnostic advice on the confidence levels of medical professionals and non-experts alike Gaube et al. (2023); van Leeuwen et al. (2021b); Tariq et al. (2020); van Leeuwen et al. (2021a); Gaube et al. (2021); Jacobs et al. (2021); Lee et al. (2019).

**LLMs in Business Applications** AI-driven systems are increasingly utilized to automate a variety of tasks, from data-driven personalization and customer experience enhancement to market

and customer prediction, dynamic pricing, and decision-making optimization (Borges et al., 2021; Gacanin & Wagner, 2019; Grewal et al., 2021; Keding, 2021). One specific focus has been applying Automated Machine Learning (AutoML) in business domains, which aims to mitigate the barrier of technical expertise by offering fully-automated solutions for model selection and hyperparameter tuning. Schmitt (2023) employed four business-oriented datasets from the UCI repository Newman et al. (1998) for evaluation. Moreover, top business consulting firms like MacKinsy&Companly are already incorporating LLMs into client solutions. Furthermore, they introduce their own generative AI solution "Lilli" for colleagues (MacKinsy&Company). Despite this, there is a notable absence of scholarly research offering analytical evaluations of LLMs' applicability in resolving business consulting cases.

