# OpenReview forum: "Hypothesis- and Structure-based prompting for medical and business diagnosis"
_ICLR.cc/2024/Conference — Submitted to ICLR 2024_

### Official Review · Reviewer_ueoZ · 2023-10-31

**Soundness:** 3 good
**Presentation:** 3 good
**Contribution:** 2 fair
**Rating:** 3
**Confidence:** 4

**Summary:**

This paper proposes the use of a MECE (Mutually Exclusive, Collectively Exhaustive) structure in the prompts of an LLM (Language Model) to assist in breaking down complex problems into hypotheses. The prompts also address the prioritization of these hypotheses, actively validating each hypothesis by requesting data from a user, and maintaining a holistic view to determine the depth of analysis. The proposed prompts yield improved results when evaluated in business and medical cases by experts.

**Strengths:**

S1. The proposed techniques for prompting, based on hypotheses and structure, make sense and have broad applicability.

**Weaknesses:**

W1. The proposed HS prompting techniques seem to be an extension of existing methods such as ToT and GoT. Yet, the authors did not include these methods as their baselines, making it hard to evaluate the effectiveness of the HS prompting.

W2. In the experiments, the criteria used to evaluate the models were not clearly defined and justified. For example, the background of the professionals and their scoring standards is not clear. It is also unclear whether the experts who set the criteria participated in the model scoring process (which could lead to potential bias).

W3. The business case study did not show a significant advantage for the proposed prompting techniques. More discussion on the cause is needed.

W4. More objective metrics should be considered in the experiments. For example, runtime, user feedback statistics, and other quantitative metrics.

W5. Presentation issues: Some data charts seem misaligned with the text. The paper requires a major revison.

**Questions:**

See the weaknesses.

---

> ### Author Response · Authors · 2023-11-21
>
> We appreciate Reviewer ueoZ's insightful feedback on our work. It is encouraging to hear that you find the proposed HS prompting techniques, grounded in hypotheses and structure, to be logical and broadly applicable.
>
> ### W1. Baseline missing
> > W1. The proposed HS prompting techniques seem to be an extension of existing methods such as ToT and GoT. Yet, the authors did not include these methods as their baselines, making it hard to evaluate the effectiveness of the HS prompting.
>
> In response to your observation about the HS method and its relation to ToT and GoT, we clarify that while there are conceptual similarities, the application context of HS is distinct. The interactive, user-driven nature of our tasks, where the model's questions and the user's responses continuously shape the problem-solving process, differs fundamentally from the more structured approach of ToT and GoT. This difference in task structure and interaction pattern influenced our decision to focus on HS without including these methods as direct baselines, as their integration presented practical challenges given the nature of our tasks
>
> ### W2, W4. Criteria clarify
> > W2. In the experiments, the criteria used to evaluate the models were not clearly defined and justified. For example, the background of the professionals and their scoring standards is not clear. It is also unclear whether the experts who set the criteria participated in the model scoring process (which could lead to potential bias).
>
>
> We appreciate your attention to the evaluation criteria and the background of the professionals involved. As detailed in Sections 4.1 and 4.2 of our paper, the business consultants evaluating our models hold MBAs or have over five years of experience from reputable consulting firms. In the medical domain, our evaluators are licensed doctors with more than five years of clinical experience, specializing in various fields including cardiology and orthopedics. The scoring standards followed by these professionals are based on the criteria outlined in Section 4. We acknowledge the importance of an unbiased evaluation process and confirm that the experts who set the criteria were not involved in the scoring process. This will be clarified in the revised version of our paper.
>
> > W4. More objective metrics should be considered in the experiments. For example, runtime, user feedback statistics, and other quantitative metrics.
>
> Thank you for raising the issue of incorporating more objective metrics into our evaluation process. We understand the importance of quantifiable measures in research. However, for the specific nature of our tasks, which focus on generating insightful questions and hypotheses to identify root causes in many-to-one problems, human expert evaluations, such as user feedback statistics and interviews, are the most appropriate and informative methods of assessment.
>
> The complexity of our tasks lies in evaluating the effectiveness of GPT-4's problem-solving approach, not merely in determining whether it arrives at the correct answer. This qualitative aspect of problem-solving, which involves understanding and evaluating the reasoning process, makes it challenging to apply traditional objective metrics. We kindly ask for your understanding regarding the difficulty in incorporating other types of objective metrics for this particular study.
>
>
>
> ### W3. In business cases, HS is not high.
> > W3. The business case study did not show a significant advantage for the proposed prompting techniques. More discussion on the cause is needed.
>
> For insights into the performance of our prompting techniques in business cases, we refer you to Figure 2 of our paper. This figure demonstrates that the HS method consistently scores higher than other baseline methods in business scenarios when averaged across different cases.
>
> ### W5. presentation issue: some charts seem misaligned with the text
> > W5. Presentation issues: Some data charts seem misaligned with the text. The paper requires a major revision.
>
> We are grateful for your feedback regarding the alignment of data charts with the text. We acknowledge the importance of a clear and accurate presentation and will undertake a major revision to ensure that all charts and text are correctly aligned and presented coherently.
>
> We are happy to answer any further questions. Thanks again for your careful assessment.

---

> > ### Comment · Reviewer_ueoZ · 2023-11-23
> > **Thanks for yor reply**
> >
> > The reviewer thanks the author for their reply. However, many of my concerns are not addressed, so I will keep my original score.

---

> > > ### Author Response · Authors · 2023-11-23
> > >
> > > To Reviewer ueoZ, we appreciate your acknowledgement of our response, even though it did not fully address your concerns. Your feedback has been invaluable, and we will consider it carefully to enhance our work.

---

### Official Review · Reviewer_hnXU · 2023-11-01

**Soundness:** 1 poor
**Presentation:** 2 fair
**Contribution:** 1 poor
**Rating:** 1
**Confidence:** 4

**Summary:**

# Summary

## What is the problem?
How can we best use LLMs in highly interactive settings with lots of back-and-forth between users to most effectively solicit information and arrive at correct conclusions given that solicited information.

## Why is it impactful?
There are many problems of this format, including medical diagnosis and business case problem discovery. Better methods to leverage LLMs for these applications would be impactful.

## Why is it technically challenging/interesting (e.g., why do naive approaches not work)?
There is a breadth of interesting related literature on how to better use LLMs in iterative reasoning problems, much of which is appropriately flagged by the authors. The existence of this prior work illustrates the challenge of this area. These problems are also challenging to evaluate properly, given that whether or not an LLM agent "reasoned correctly" and "most efficiently" is often ill-posed and challenging to formalize.

## Why have existing approaches failed?
The authors allege that existing approaches fail in certain circumstances that (they implicitly argue) are essential in healthcare or business use cases. However, these challenges are not quantified nor sufficiently justified by their empirical results. Further, they fail to sufficiently comment on related resources, such as https://jamanetwork.com/journals/jama/article-abstract/2806457 this paper which examines medical diagnostic performance in a set of published clinical case report challenges and find generally positive results in that setting.

## What is this paper's contribution?
This paper proposes a prompting strategy to perform these iterative, chat-based tasks.

## How do these methods compare to prior works?
They compare to IO (which is never clearly defined), CoT, and two variants therein that rely on adding a "ask for more information once" modifier to the prompts.

## How do they validate their contributions?
They perform a set of quantitative vignettes of their model's performance on 3 business case studies and 4 medical diagnosis challenges, evaluated by human experts.

**Strengths:**

## Key Strengths (reasons I would advocate this paper be accepted)
  1. This is an important an interesting challenge.

**Weaknesses:**

## Key Weaknesses (reasons I would advocate this paper be rejected)
  1. You assess your method on only 3 business cases for business consulting. This is dramatically too few examples on which to base any generalizable conclusion for the performance of this method. You need to assess this method on a much larger set of business consulting problems in order to argue that your approach has merit over alternative approaches in a reliable manner that should be expected to generalize across a meaningful subset of business consulting problems.
  2. Similarly for medical cases, you need to experiment with more than 4 cases (I know you started with 5, but one ended up being excluded).
  3. The quoted justification below for rejecting baselines feels insufficient. Your framework is also repetitive (in that it is recursive) to arrive at a single end point, and the fact that use ChatGPT rather than the API does not invalidate studies that require API use, as you could (and should, for robustness in your experiments) be able to implement your study using a simple program that leverages the API. Quoted justification: "While these methods utilize the GPT-4 API to integrate tree or graph search algorithms, they’re not directly adaptable to our chat interface, where humans interact with the model. Furthermore, these methods best suit tasks that can be broken down into repetitive steps, with a clearly defined endpoint, while business and medical diagnosis tasks are not."
  4. IO is not clearly defined in your work. CoT is defined as an acronym, but you don't explicitly indicate what exact prompting strategy is used and how it differs from your approach to justify your experiments.
  5. A key part of the challenge here is one of evaluation; which evaluation metrics should be used, how can they be efficiently assessed at scale, what factors of the input motivate success or failure on different metrics, etc. You do not offer any significant commentary on these challenges nor offer solutions for them, which significantly undercuts your impact here.
  6. As you are using human evaluators, you need to state that you have appropriate IRB approval to run this study (in order to solicit the survey responses from your human evaluators).

**Questions:**

Unfortunately I do not foresee any changes that could motivate me to change my review at this time. You would need to fully re-do your evaluation and experiments at a much greater scale for me to consider a change in score here.

**Details Of Ethics Concerns:**

They perform a human subjects study in this work (by leveraging human evaluations) but do not assert appropriate IRB approval or ethics review to perform such a study.

---

> ### Author Response · Authors · 2023-11-21
>
> Thank you, Reviewer hnXU, for recognizing the importance and intrigue of the challenges addressed in our study. Your acknowledgement is greatly valued and motivates our ongoing research in this area
>
> ### Intro. “Existing method fails in …” is not true [JAMA]
> > The authors allege that existing approaches fail in certain circumstances that (they implicitly argue) are essential in healthcare or business use cases. However, these challenges are not quantified nor sufficiently justified by their empirical results. Further, they fail to sufficiently comment on related resources, such as https://jamanetwork.com/journals/jama/article-abstract/2806457 this paper examines medical diagnostic performance in a set of published clinical case report challenges and finds generally positive results in that setting.
>
> Thank you for referencing the JAMA paper. We acknowledge that the problem setting in that study differs significantly from ours. In their setup, all patient conditions are presented at once for the model to make predictions. Conversely, our approach with GPT-4 involves asking pertinent questions to gather data, which is crucial for quality analysis and responses. Furthermore, their methodology doesn't utilize common methods like CoT or IO. Instead, they provide detailed instructions specific to their test cases, but do not formally formulate the prompt structure. Our HS method, on the other hand, establishes a standardized prompt format applicable across diverse domains, including medical and business scenarios. This adaptability and structure are what we aim to emphasize as our key contributions.
>
> ### W1,2,3. Few cases, Baseline
> We appreciate your critique of our justification for not using certain baseline methods. While our framework does involve a recursive approach, the nature of our tasks, which involve dynamic and user-responsive problem-solving, does not align well with the predetermined search patterns of methods that require API usage. The key difference lies in the unpredictability and flexibility of the problem-solving steps in our tasks, which contrasts with the more defined and repetitive structures suited to API-driven methods. Our decision to prioritize the HS method was based on its compatibility with the evolving and interactive nature of our study's tasks.
>
>
> ### W4. IO, CoT definition
> > IO is not clearly defined in your work. CoT is defined as an acronym, but you don't explicitly indicate what exact prompting strategy is used and how it differs from your approach to justify your experiments.
>
> Regarding the definitions of IO (input-output method) and CoT (Chain-of-Thought method), we adhere to the definitions provided in Wei et al., 2022, as cited in our paper. We believe these established definitions accurately represent the methodologies we reference and compare against in our work.
>
> Wei, Jason, et al. "Chain-of-thought prompting elicits reasoning in large language models." Advances in Neural Information Processing Systems 35 (2022): 24824-24837.
>
> ### W5. Evaluation metric
> > A key part of the challenge here is one of evaluation; which evaluation metrics should be used, how can they be efficiently assessed at scale, what factors of the input motivate success or failure on different metrics, etc. You do not offer any significant commentary on these challenges nor offer solutions for them, which significantly undercuts your impact here.
>
> Addressing the challenge of evaluation, we recognize that in business consulting and medical diagnosis, standard evaluation metrics are not always clearly defined. Following the approach of Singhal et al., 2023, we employed human experts to develop criteria tailored to our objectives. Our methodology involved starting with widely recognized criteria and then refining these with domain experts to focus on evaluating GPT-4's ability to generate viable hypotheses and ask relevant questions. This two-step process ensures that our criteria are both relevant and manageable for evaluators.
>
> ### W6. IRB approval
> > As you are using human evaluators, you need to state that you have appropriate IRB approval to run this study (in order to solicit the survey responses from your human evaluators).
>
> We appreciate your concern regarding ethical approval for using human evaluators. As stated in our paper, we have obtained the necessary approval for conducting human-related experiments from the Cambridge Engineering Ethical approval board. This process is detailed here:  https://www.researchandfinance.eng.cam.ac.uk/ethics. It's important to note that IRB approval is typically required for biomedical research, which differs from the scope of our study.
>
>
> We are happy to engage in further discussion to clarify any unresolved concerns. Thanks again for your detailed review.

---

> > ### Comment · Reviewer_hnXU · 2023-11-22
> > **Thank you for your comments**
> >
> > Thank you for providing follow-up comments. Unfortunately, my score has not changed, as the largest concern I have is the very small number of case studies used to evaluate this model (3 for business consulting, 4 for healthcare). Thank you nonetheless for your time and energy.

---

> > > ### Author Response · Authors · 2023-11-22
> > >
> > > Thank you, reviewer hnXU, for your continued engagement and feedback. We understand your concern regarding the limited number of case studies in our research. We wish to emphasize that conducting validations in complex domains such as business consulting and medical diagnosis is both financially and temporally demanding. As outlined in our ethics statement section, the compensation for participants assessing these case studies was significant, resulting in a total expenditure of £600 for the survey. This budget constraint played a key role in limiting the number of cases we could feasibly include. Despite these limitations, we carefully selected the most representative cases for each domain to ensure the quality and relevance of our findings. We hope that our preliminary results will serve as a foundation for future research, allowing others to build upon and expand our work

---

> > > > ### Comment · Reviewer_hnXU · 2023-11-23
> > > > **Further explanation of the issue with too few cases**
> > > >
> > > > I appreciate your concerns about evaluation cost and the importance of preliminary studies. But, to put my concerns in greater context, consider your business consulting setting. In this setting, you assess your model vs. the 3 baselines on 3 cases. Your model takes first place on 2 of those three cases.
> > > >
> > > > If all of the tested models are equally likely to take first on a given case, then the probability of observing your model taking first on any one case would be (1/4). Thus, in this hypothetical scenario, the probability of observing your model take first place on 2 of three possible samples would be $3\cdot\left(\frac{1}{4}\right)^2\cdot\left(\frac{3}{4}\right)^1\approx0.14$. this implies that there is a nearly 15% chance that we would see results that look as good as yours do just due to random chance alone. This is significantly higher than the 5% or even 1% cutoff that is normally used to judge when results should be interpreted as reliable, as opposed to being seen as statistically insignificant.
> > > >
> > > > In other words, without expanding your set of cases, even if every other aspect of your study were perfect, you could still never argue that your model should be seen to improve over existing baselines with a significance threshold below 14%, which really limits the publishability of this work.

---

> > > > > ### Author Response · Authors · 2023-11-23
> > > > >
> > > > > To Reviewer hnXU, we acknowledge your valid points regarding the statistical significance of our study's results. We will aim to expand upon these in future research to provide more statistically robust evidence. We thank you once again for your valuable time and insights.

---

### Official Review · Reviewer_HT4V · 2023-11-01

**Soundness:** 2 fair
**Presentation:** 3 good
**Contribution:** 2 fair
**Rating:** 3
**Confidence:** 3

**Summary:**

This paper introduces a novel prompting strategy, HS, which starts by breaking down the problem space using the concept of Mutually Exclusive and Collectively Exhaustive (MECE), and it proceeds to prioritize and validate hypotheses through interaction with the user. The paper also introduces easy-to-follow guidelines for crafting examples for potential users of HS prompts. Experiments on business consulting and medical diagnosis in 'many-to-one' scenarios show that HS prompting outperforms previous prompting strategies, even when modified for these particular tasks, indicating potential applicability for LLMs in challenging, real, domain-specialized scenarios.

**Strengths:**

- The motivation of the paper is ambitious and addresses real-world problems using LLMs.
- The experiments conducted cover both the business and medical domain, which can potentially demonstrate the impact of the paper.
- The proposed prompting is well-thought-out and convincing when using GPT-4.

**Weaknesses:**

- The method heavily relies on GPT-4's ability, which may limit the applicability of the approach to other LLMs. CoT, ToT, and GoT are general methods that can be used in any other LLMs regardless of how much knowledge is stored in LLMs (they act as an aid to LLM reasoners). However, the assumption of HS is that LLMs are very knowledgeable about defining any problem landscape and are great at generating possible hypotheses, which is not just the role of a reasoner but of an oracle-knowledge base and reasoner at the same time). To show the broad applicability of the approach, the authors should use other LLMs to demonstrate the effectiveness of HS (possibly using a retriever if some LLMs do not hold enough knowledge).
- The proposed method resembles a human-AI interactive version of ToT or GoT. The idea itself is very practical and useful, but the credibility of this approach depends on the user.

**Questions:**

- How is the citation (Zheng et al., 2023) for neural architecture search related to real-world modeling? Also, I think 'general tasks' should be 'general NLP tasks'.
- How can we guarantee that the options that LLMs offer are MECE if used by a non-expert?
- Are there citations or backup arguments for the claims below?
1. 'LLMs need to apply this knowledge in a structured and efficient manner, especially when solving many-to-one problems'
2. 'While CoT excels in one-to-one mapping problems, it falters when multiple potential root causes must be explored'

---

> ### Author Response · Authors · 2023-11-21
>
> We are grateful for Reviewer HT4V's thoughtful evaluation of our work. Your acknowledgement of the paper's ambitious motivation and its application to real-world problems using LLMs is greatly appreciated. We also value your recognition of our experiments covering business and medical domains, as well as your positive view of the well-thought-out nature of our proposed prompting method with GPT-4.
>
> ### W1. Other LLM: CoT, ToT… used in any LLMs regardless of how much knowledge is stored in LLMs
> > The method heavily relies on GPT-4's ability, which may limit the applicability of the approach to other LLMs. ... However, the assumption of HS is that LLMs are very knowledgeable about defining any problem landscape and are great at generating possible hypotheses, which is not just the role of a reasoner but of an oracle-knowledge base and reasoner at the same time). ...
>
> We appreciate your concern regarding HS's reliance on GPT-4's capabilities. While HS does assume that LLMs possess a degree of logical reasoning, it does not necessitate extensive domain-specific knowledge. The HS framework is designed to guide LLMs in enhancing their domain understanding by prompting them to ask relevant questions, even with limited initial knowledge. For instance, in business scenarios like declining sales, the model, with basic knowledge (e.g., declining sales is due to either new competitors or decreasing product quality), can formulate hypotheses and ask pertinent questions across various domains such as automotive, food, or oil industries. Similarly, in medical contexts, it might start by determining the urgency or nature of a case. This methodology can be adapted to LLMs with varying knowledge capacities. We acknowledge the value of testing HS with less powerful LLMs than GPT-4 and aim to pursue this in future research.
>
> ### W2, Q2. The credibility of this approach depends on the user
> > How can we guarantee that the options that LLMs offer are MECE if used by a non-expert?
>
> > The proposed method resembles a human-AI interactive version of ToT or GoT. The idea itself is very practical and useful, but the credibility of this approach depends on the user.
>
> Your question about the user's expertise is vital. We want to clarify that HS targets junior experts in business and healthcare, assisting them in identifying root causes with logical, explainable flows. However, the framework is also designed to be accessible to non-experts. In our experiments, we limited the user's role to providing data as requested by GPT-4. The prompts, once generated for a specific domain, can be reused by non-expert users for multiple questions within that domain. This approach, coupled with clear instructions, makes HS more conducive to generating quality examples, surpassing other prompting methods in terms of user guidance.
>
> ### Q1. How … neural architecture search related to real-world modelling
> > How is the citation (Zheng et al., 2023) for neural architecture search related to real-world modelling? Also, I think 'general tasks' should be 'general NLP tasks'.
>
> Regarding the citation of Zheng et al., 2023, in the context of neural architecture search, it exemplifies how LLMs utilize domain-specific knowledge (in this case, deep learning model design) to propose optimized neural architectures. LLMs draw on their understanding of the relationship between training loss curves and model complexity to suggest adjustments in model size. This process is indicative of the applicability of LLMs in real-world modelling scenarios. Also, we agree with your suggestion to specify 'general NLP tasks' for clarity and precision.
>
> Zheng, Mingkai, et al. "Can GPT-4 Perform Neural Architecture Search?." arXiv preprint arXiv:2304.10970 (2023).
>
> ### Q3. Citations or backup
> > Are there citations or backup arguments for the claims below?
>
> Thank you for pointing out the missing citations.
>
> 'LLMs need to apply this knowledge in a structured and efficient manner, especially when solving many-to-one problems': Many papers and textbooks emphasise the role of hypothesis-based and structure-based thinking in business.
> Eisenmann, Thomas R., Eric Ries, and Sarah Dillard. "Hypothesis-driven entrepreneurship: The lean startup." Harvard business school entrepreneurial management case 812-095 (2012).
>
> 'While CoT excels in one-to-one mapping problems, it falters when multiple potential root causes must be explored': We empirically observe the falters of CoT in many-to-one problems, which is the motivation for this research. You can find it in our results.
>
> We thank the reviewer once again for their thorough assessment and are more than happy to engage further.

---

> > ### Comment · Reviewer_HT4V · 2023-11-23
> > **Answer to authors**
> >
> > Thank you for taking the time to address my questions. I am still not convinced by many of the concerns I have raised, particularly regarding the method's reliance on GPT-4's extensive domain knowledge and the user's expertise. To make the paper more convincing, the authors should consider conducting experiments to demonstrate: 1) the reliability of using other LLMs for HS and 2) experiments involving users with varying levels of domain expertise.

---

> > > ### Author Response · Authors · 2023-11-23
> > >
> > > To Reviewer HT4V, we are grateful for the time you took to review our responses. We understand your concerns about the method's reliance on GPT-4 and the level of user expertise. Your suggestions regarding additional experiments with other LLMs and users of varying expertise are insightful and will be a crucial part of our future research efforts.

---

### Official Review · Reviewer_y4hX · 2023-11-06

**Soundness:** 3 good
**Presentation:** 2 fair
**Contribution:** 2 fair
**Rating:** 3
**Confidence:** 2

**Summary:**

This paper introduces a novel method called Hypothesis- and Structure-based Prompting (HS) for enhancing the problem-solving capabilities of Large Language Models (LLMs) in healthcare and business. The approach breaks down the problem space using a Mutually Exclusive and Collectively Exhaustive (MECE) framework, enabling LLMs to generate, prioritize, and validate hypotheses through targeted questioning and data collection. The paper provides an easy-to-follow guide for crafting examples, allowing users to develop tailored HS prompts for specific tasks.

**Strengths:**

- The paper found that adding a single sentence about requesting data improved the performance of HS prompting effectively, similar to a previous study. This finding demonstrates the effectiveness of the HS approach and its potential for further improvement.
- This approach breaks down the problem space using a Mutually Exclusive and Collectively Exhaustive (MECE) framework, which is a unique and effective way of approaching complex problems.
- The paper provides an easy-to-follow guide for generating examples, allowing users to create appropriate examples tailored to their specific tasks. By aligning the examples with the structure-based and hypothesis-based approach, users can stimulate the LLMs to solve problems more effectively and efficiently.

**Weaknesses:**

- Limited comparison to existing methods: While the paper enlists domain experts to validate the HS method and provide a comparison to existing baseline methods, the comparison is limited to a few specific methods. It would be beneficial to see a more comprehensive comparison to a wider range of existing methods.
- The paper focuses on the application of the HS method to healthcare and business diagnosis, and it is unclear how generalizable the approach is to other domains or problem types.
- The paper's qualitative evaluation is limited to a few cases with a panel of consultants or medical doctors. Performance consistency is a major concern with such limited of observed samples.

**Questions:**

- Can the authors provide more real-world case studies where the HS method was used successfully in healthcare or business? This would help to demonstrate the practical applicability of the approach and provide more evidence of its effectiveness.
- How generalizable is the HS method to other domains or problem types? Have you considered applying the approach to other areas, such as finance or engineering? If so, what were the results?
- Can you provide more details on the process of generating examples for the HS method? How do you ensure that the examples are of high quality and representative of the problem space?
- How does the HS method compare to other approaches that incorporate human-in-the-loop feedback or other forms of human-machine collaboration? Have you considered the potential benefits and drawbacks of these approaches?
- Can you provide a more comprehensive comparison to a wider range of existing methods? This would help to demonstrate the relative strengths and weaknesses of the HS method compared to other approaches.

---

> ### Author Response · Authors · 2023-11-21
>
> We thank Reviewer y4hX for their insightful feedback. We are particularly encouraged by your recognition of our approach in improving HS prompting effectiveness and the novel application of the MECE framework. Your appreciation of our guide for generating tailored examples also reinforces the practical value of our work.
>
> ### Q1, 2.More cases and domains
> > Can the authors provide more real-world case studies where the HS method was used successfully in healthcare or business? …
>
> Thank you for the feedback regarding the application of the HS method across various real-world scenarios. Regarding the inclusion of more case studies in healthcare and business, as mentioned in our common response, our selection was limited to three business consulting cases and four medical cases, primarily due to budget constraints for expert evaluations. These cases were chosen carefully to represent diverse scenarios within business consulting and medical diagnostics, specifically focusing on chest pain diagnosis in healthcare.
>
> > How generalizable is the HS method to other domains or problem types? …
>
> We acknowledge the potential applicability of the HS method in other fields such as finance and engineering. However, these domains fall outside the scope of our current research. Our primary focus is on addressing many-to-one problems, particularly those involving the identification of a primary cause from multiple symptoms or observations. This focus guided our choice of business consulting and medical diagnostics as the primary application areas for this study.
>
> In future work, we aim to explore the extension of the HS method to additional domains and further evaluate its effectiveness across a broader range of applications. This expansion will enable us to better understand the generalizability and utility of the HS method in various real-world contexts.
>
> ### Q3.Details of generating examples for HS.
> > Can you provide more details on the process of generating examples for the HS method? How do you ensure that the examples are of high quality and representative of the problem space?
>
> Thank you for your inquiry about the example generation process in the HS method. As outlined in Section 3 of our paper, the HS method includes an easy-to-follow guide that encourages users to generate and include examples in the prompt. We start by providing general HS problem-solving instructions that are applicable across various domains. Users are then guided to bring in an example from a specific domain and structure it according to each instruction, mirroring the method we used to generate our own examples.
>
> We recognize that the effectiveness of the HS method might diminish if users input low-quality examples. This is a challenge inherent in most prompting methods, not exclusive to HS. However, we believe that HS, with its structured guidance, better assists users in creating examples of acceptable quality compared to other prompting techniques.
>
> ### Q4. Comparing Human-in-the-loop methods
> > How does the HS method compare to other approaches that incorporate human-in-the-loop feedback or other forms of human-machine collaboration? Have you considered the potential benefits and drawbacks of these approaches?
>
> Your question regarding the comparison of the HS method with human-in-the-loop and other human-machine collaborative approaches is a compelling direction for consideration. The HS method is distinct in that once a prompt is tailored for a specific domain, it can be reused for multiple queries within that domain, even by non-expert users. This reuse contrasts with human-in-the-loop methods, where the quality of output heavily depends on the user’s expertise and experience level. While human-in-the-loop approaches have their merits, the HS method offers more consistency in quality, especially for users lacking deep domain expertise.
>
> ### Q5. Baselines
> > Can you provide a more comprehensive comparison to a wider range of existing methods? This would help to demonstrate the relative strengths and weaknesses of the HS method compared to other approaches.
>
> Thank you for suggesting a broader comparison with existing methods. As we answer in the common response, we acknowledge the importance of such comparative analysis. However, our decision to not include methods like ToT and GoT as baselines were influenced by the challenges in integrating their structured search algorithms into our unique task design. Our HS method focuses on an interactive model where the user's input dynamically shapes the conversation, which differs significantly from the more static and predefined pathways of these baseline methods. Thus, while we understand the value of a comprehensive comparison, the distinct nature of our interactive tasks makes such a direct comparison challenging.
>
> We thank the reviewer once again for encouraging feedback and careful assessment.

---

### Author Response · Authors · 2023-11-21
**Common response**

We are elated to receive detailed feedback from the reviewers. We are very grateful for their time and efforts. We are glad that the reviewers found the motivation to solve real-world many-to-one problems interesting. We addressed specific questions from each reviewer in our response and addressed the common issue.

### Q1. Baselines (y4hX Q5; hnXU W3; ueoZ W1)
In addressing the reviewers' concerns about the absence of certain baseline methods like ToT, GoT, and RAP in our study, it's crucial to understand the unique challenges of integrating these methods into our specific task structure. Our research involves interactive scenarios where the user provides data in response to the model's queries. This dynamic interaction complicates the direct application of baseline methods that rely on predefined search algorithms asking GPT-4 for options and making selections. The non-linear and undefined nature of steps in our tasks makes it challenging to determine how and when to effectively integrate search algorithms outside LLMs in these baselines. This difficulty in application, rather than an oversight, influenced our decision to focus on developing and testing the HS method, which is better suited to the flexible and evolving interaction pattern of our study.


### Q2. More cases/domain (y4hX Q1,Q2; hnXU W1,W2)
Our research included a focused selection of three business consulting cases and four medical cases, chosen due to budget constraints for expert evaluations and to represent a diverse array of scenarios within these domains. While we acknowledge the potential of the HS method in other areas such as finance and engineering, our current focus is on many-to-one problems in business consulting and medical diagnostics. We aim to extend our research to additional domains in the future to further explore and demonstrate the HS method's versatility and effectiveness.

### Q3. Criterion (hnXU W5;  ueoZ W2)
In terms of evaluation, we employed a two-step process involving widely recognized criteria, which were then refined by domain experts. This approach was necessary due to the qualitative nature of our tasks, focusing on GPT-4’s problem-solving process rather than just its ability to arrive at correct answers. We understand the importance of objective metrics but believe that for our specific tasks, expert evaluations, user feedback statistics, and interviews are the most relevant and informative methods of assessment. We acknowledge the challenges in applying traditional objective metrics to our study and seek understanding in this regard.

### Q4. Credibility on Users(y4hX W3; HT4V W2, Q2)
Regarding user interaction with the HS method, we emphasize that it is designed for junior experts in business and healthcare, aiding them in logical problem-solving. Nonetheless, it is also accessible to non-experts. Our methodology, detailed in Section 3 of our paper, includes an easy-to-follow guide for users to generate and include examples in their prompts, thereby ensuring the relevance and quality of these examples. We recognize that the quality of user-generated examples is critical and believe that HS's structured approach is more effective in guiding users to create better quality examples than other prompting methods.

We are happy to answer any other unresolved concerns. We thank the reviewers once again for their thoughtful assessment of our work.

---

### Meta-Review · Area_Chair_3gkJ · 2023-12-06

**Metareview:**

This paper proposes an interactive, iterative prompting strategy which exploits properties of certain output spaces (namely mutual exclusivity in many-to-one-mappings). The motivation here are medical and business applications. When combined with GPT-4, the authors report promising results using this strategy on a handful of datasets and tasks.

While reviewers liked the direction of interactive prompting, there was general agreement that the contribution here is limited, due in large part to its total reliance on GPT-4. It is unclear if other LLMs would benefit from the proposed strategy, which great reduces potential impact (the authors responded to this critique in rebuttal, but defer this to "future work"). Another weakness of the work is the limited evaluation in terms of datasets, and evaluation metrics more generally. Ultimately, as is this work provides evidence that rather general interactive prompting strategy may yield some benefits on a few tasks when using GPT-4; this of limited value to the broader community.

**Justification For Why Not Higher Score:**

The work provides limited insights, given its empirical setup and reliance on GPT-4 (only).

**Justification For Why Not Lower Score:**

N/A

---

### Decision · Program_Chairs · 2024-01-16

Reject